# Fermentation Enhanced Biotransformation of Compounds in the Kernel of *Chrysophyllum albidum*

**DOI:** 10.3390/molecules25246021

**Published:** 2020-12-19

**Authors:** Oluwatofunmi E. Odutayo, Emmanuel A. Omonigbehin, Tolulope D. Olawole, Olubanke O. Ogunlana, Israel S. Afolabi

**Affiliations:** 1Biochemistry Department, College of Science and Technology, Covenant University, Ota 100122, Nigeria; oluwatofunmi.obaseki@gmail.com (O.E.O.); tolulope.olawole@covenantuniversity.edu.ng (T.D.O.); banke.ogunlana@covenantuniversity.edu.ng (O.O.O.); 2Molecular Biology Laboratory, College of Science and Technology, Covenant University, Ota 100122, Nigeria; eaomonigbehin@yahoo.com

**Keywords:** *Chrysophyllum albidum*, seed, kernels, food, processing, fermentation, bioactive

## Abstract

*Chrysophyllum albidum* Linn (African star apple) is a fruit with extensive nutritional and medicinal benefits. The fruit and kernel in the seed are both edible. Strains of lactic acid bacteria (LAB) were isolated from fermented seeds and assessed for probiotic characteristics. The extracts in both the unfermented and the fermented aqueous extracts from the kernels obtained from the seeds of *C. albidum* were subjected to analysis using the gas chromatography/mass spectrometry (GC-MS) method. This analysis identified the bioactive compounds present as possible substrate(s) for the associated organisms inducing the fermentation and the resultant biotransformed products formed. Three potential probiotic LAB strains identified as *Lactococcus raffinolactis* (ProbtA1)*, Lactococcus lactis* (ProbtA2a)*,* and *Pediococcus pentosaceus* (ProbtA2b) were isolated from the fermented *C. albidum* seeds. All strains were non hemolytic, which indicated their safety, Probt (A1, A2a, and A2b) grew in an acidic environment (pH 3.5) during the 48-h incubation time, and all three strains grew in 1% bile, and exhibited good hydrophobicity and auto-aggregation properties. Mucin binding proteins was not detected in any strain, and bile salt hydrolase was detected in all the strains. l-lactic acid (28.57%), norharman (5.07%), formyl 7*E*-hexadecenoate (1.73%), and indole (1.51%) were the four major constituents of the fermented kernel of the *C. albidum*, while 2,5-dimethylpyrazine (C1, 1.27%), 3,5-dihydroxy-6-methyl-2,3-dihydropyran-4-one (C2, 2.90%), indole (C3, 1.31%), norharman (C4, 3.01%), and methyl petroselinate (C5, 4.33%) were the five major constituents of the unfermented kernels. The isolated LAB are safe for consumption. The fermenting process metabolized C1, C2, and C5, which are possible starter cultures for the growth of probiotics. Fermentation is an essential tool for bioengineering molecules in foods into safe and health beneficial products.

## 1. Introduction

The seeds of plants are good sources of food for humans and animals because of their wealth of nutrients. However, not all seeds are edible. These non-edible seeds may have the potential to be transformed into useful purpose(s). Efforts are on the increase to reveal their potential and the benefits of some of these non-edible seeds since they may be transformed into useful purpose(s). Fermentation also transforms the physical properties of the seeds, leading to other products from the process, while at the same time transforming the natural phytochemicals. This is evident by the change in color, texture, and some other physical properties in products (liquid pap from cereals like sorghum, nutritious milk from the seeds of soybeans or *Adenanthera pavonina*, flakes and tapioca from cassava tubers, food condiments like iru or maggi from African locust beans) derived from such fermentation processes [1,2,3,4,5,6,7]. The study on under-utilized and non-utilized seeds is of increasing global importance as the quest for food security increases. *Chrysophyllum albidum* Linn, which is also known as the African star apple, is commonly called “agbalumo” by the Yoruba tribe or “Udara” by the Igbos tribe in Nigeria. It is a forest tree species belonging to the family Sapotaceae. The seed, which has a tasty pulp and about three to five seeds within the fruit, is found in tropical countries of Africa. There are white kernels are present in the hard, dark brown in color, and shiny seeds when broken. These kernels have a high level of fat and carbohydrate, making it a good source of energy [8]. It could also be a good source of calcium, potassium, and magnesium as well as protein [9]. However, despite availing reports on the nutritional qualities of this seed, it is still very under-utilized and is either used for local games or discarded after consuming the pulp.

Raw sources of food contain phytochemical compounds that undergo bio-transformation to other forms of bioactive compounds by fermenting microorganisms. The phytochemical compounds are components of food that influence physiological or cellular activities in the animals or humans that consume them [5]. They are present in small quantities in foods such as fruits, vegetables, and whole grains. They also provide health benefits beyond the primary nutritional value [10,11]. Phytochemicals present in different plants modulate microbial activity, and different groups of phytochemical compounds can impact microbial composition at various levels [12,13]. These compounds in plants can also serve as carbon sources for these organisms. An example is the release of organic acids such as citric and fumaric acids by tomato roots, which have been reportedly used to culture beneficial microbes like the plant growth-promoting *Rhizobacteria* [12,14].

For over a century now, fermentation of foods has been used as a tool used by humans in food processing [3]. The consumers mostly prefer fermented foods due to their enhanced level of taste, nutrients, safety, shelf-life, and the ability to be digested [7,15]. Other reasons for which fermentation is still gaining popularity include the increased availability of vitamins and other biomolecules resulting from the process of fermentation [3]. Fermentation is a metabolic process whereby complex molecules are broken down into simpler derivatives as a result of the activities of microorganisms. Hence, the modern science of biotechnology and bioengineering now employ their use to achieve desirable products. Fermentation also facilitates food preservation, and the reduction of anti-nutritional factors that results in nutritional enhancement [5,16]. The observed changes are mainly as a result of the metabolic action of fermenting microorganisms that performs the transformation of natural metabolites in the food substrate.

The activities of the microorganisms that thrive under the fermenting condition often facilitate metabolic reactions that biotransform a particular compound to another. The biotransformation process may further modify food properties such as enhanced nutritional qualities, reduction in toxins and antinutrients, digestibility, palatability, shelf-life, sensory, and safety properties that are mostly desired by consumers [17]. The most common fermentation process is the natural submerged fermentation of maize, millet, and sorghum often practiced in homes for pap production. The change in the form such as taste, aroma, among others, that occurs after fermentation, can be credited to the fermentation organisms [18]. There is usually a major demand for carbon in microbial metabolism since the skeleton of all the major cellular molecules except water contains carbon atoms. Hydrogen and oxygen are, in most cases, derived from substrates such as sugars, but as a matter of fact, they can also be obtained from water [19]. However, fermentation is considered to be vastly applied to an anaerobic process, whereby organisms do not make use of molecular oxygen for respiration. It is noteworthy that even organisms that perform metabolism in this way may generally require a source of this carbon element [19]. In the natural (spontaneous) fermentation of foods, lactic acid bacteria form part of the major fermenting microorganisms, hence the reason why probiotic bacteria can be obtained from fermented sources and made commercially available.

Additionally, natural fermentation provides an opportunity for a better overview of the biochemical and microbial activities during the domestic preparation of fermented foods. On the other hand, the limitation of this natural or spontaneous type of fermentation is the high risk of spoiling the microbial communities of foods, and the domination of microbial strains that could be dangerous for human health [20]. The microbes that confer beneficial health effects to their host when administered in adequate amounts are the probiotics [21]. Notable among the probiotics are the lactic acid bacteria that have been commonly utilized for several decades in the food industry [22]. Efforts are on the increase to isolate them from different sources and commercialize these lactic acid bacteria that are a dominant fermenting microorganism involved in the spontaneous fermentation of foods. The processing of the kernels of *C. albidum* with the aid of fermentation is yet to be reported. In this study, we employed gas chromatography-mass spectrometry (GC-MS) analysis to analyze the phytochemical compounds present in the raw and the fermented kernel of *C. albidum*. These efforts should further provide understanding of the nature of the biotransformation of such identified molecules during the fermentation, and the possible health benefits of the product of the fermentation.

## 2. Results

### 2.1. Identification of the Isolated Organisms Facilitating the Fermentation

Three (3) lactic acid bacteria (LAB) isolates were obtained from the fermented *C. albidum* seeds. The isolates were named as follows: ProbtA1, ProbtA2a, and ProbtA2b. The three isolates were Gram-positive and catalase-negative. From the result of the biochemical identification, Probt (A1, A2a, and A2b) were identified as *Lactococcus raffinolactis*, *Lactococcus lactis*, and *Pediococcus pentosaceus* respectively (Figure 1).

### 2.2. Biotransformation of Molecules by the Fermenting Organisms

To the best of our knowledge, this is the first report on the bioactive composition of the unfermented (Appendix A) and the naturally fermented (Appendix A) aqueous extracts from the kernels of *C. albidum*. 2,5-Dimethylpyrazine (C1, 1.27%), 3,5-dihydroxy-6-methyl-2,3-dihydropyran-4-one (C2, 2.90%), indole (C3, 1.31%), norharman (C4, 3.01%), and methyl petroselinate (C5, 4.33%) were the five major constituents of the unfermented kernel of the *C. albidum* (Table 1). Additionally, a total of thirty-nine (39) fewer compounds were detected in the fermented aqueous extract from the kernel of *C. albidium* (Appendix A). l-Lactic acid (28.57%), norharman (5.07%), formyl 7*E*-hexadecenoate (1.73%), and indole (1.51%) were the four major constituents of the fermented kernel of the *C. albidum* (Table 2). The proposed transformation mechanisms for the major molecules identified in the unfermented kernels of *C. albidum* during fermentation are indicated in Figure 2.

## 3. Discussion

### 3.1. Identification of the Isolated Organisms Facilitating the Fermentation

Gram-positive and catalase-negative reactions are preliminary ways to identify LAB. LAB has been reported to be Gram-positive and catalase-negative (Ismail, Yulvizar & Mazhitov, 2018). In the biochemical profiling of the isolates with the API 50 CHL system (for the identification of lactic acid bacteria) shown in Table 3, the positive results obtained for each of the forty-nine (49) substrates (sugars) demonstrated the ability of the LAB isolates to utilize the enzymatic activities, related to the fermentation of the carbohydrates. LAB is known to produce lactic acid as the end result of carbohydrate fermentation. Therefore, the lactic acid produced caused a drop in pH, resulting in the positive color change for any of the fermented sugar [23].

### 3.2. Biotransformation of Molecules by the Fermenting Organisms

This fermentation process did not change the indole and the norharman (Figure 2). This finding may imply that the functional groups of these compounds are resistant to the change in acidic conditions that mostly led to phytochemical modifications during the fermentation process. The fermenting process metabolized C1, C2, and C5 (Figure 2). This study suggests that the molecules were consumed by the supporting probiotics during the fermentation of kernels from seeds of *C. albidum* (Figure 2). Hence, the three molecules are possible starter cultures for the growth of probiotics. The fermentation of the *C. albidium* also created two new molecules (Figure 2). These molecules are suspected to be the by-products of the metabolism due to the probiotics facilitating the fermentation [13]. This finding is in agreement with the reports of Goswam et al. [17] and Rezac et al. [18] stating that compounds of interest with more bioavailability can be obtained from existing foods depending on the kind of fermentation used for the processing. 2,5-Dimethylpyrazine (C1), 3,5-dihydroxy-6-methyl-2,3-dihydro-4*H*-pyran-4-one (C2), indole (C3), norharman (C4), and the methyl petroselinate (C5) were the predominant molecules in the unfermented *C. albidum*. Both identified metabolites (C1 and C2) in the unfermented *C. albidum* seeds are associated with the Maillard reaction and are of great benefit to the food industries due to their pleasant sensory qualities [24,25]. Both C1 and C2 are present in cakes, Tartary buckwheat tea, toasted oak wood, garlic oil, cane brown sugar, cocoa, and cocoa containing products [26,27,28,29,30].

Norharman (C4) is present in foods. It possesses a cytotoxic activity to cancer and Parkinson’s diseases. It is useful in controlling xenobiotics and is not mutagenic. It has antimicrobial properties against *Staphylococcus aureus*, *Bacillus subtilis*, and the plant pathogen *Agrobacterium tumefaciens* [31,32]. This compound is in normal body constituents formed endogenously [32,33]. Norharman, which is endogenously derived from the pyrolysis of proteins and amino acids, is structurally related to the nonpolar heterocyclic aromatic amines. It consists basically of pyrrole, benzene, and pyridine rings [34]. The presence of ethanol, amines, amino acids, and acetaldehyde often influences the formation of norharman. Norharman is formed endogenously by cyclization of tryptamine with a carbon unit transferred from 5-methyltetrahydrofolate [35]. Indole (C3) is naturally derived from norharman depending on the level of acidity of the medium or food [36,37,38].

Ethanol (I-2) is a primary product of alcoholic fermentation [39]. The production of a proposed intermediate (I-4) for this fermentation was also previously implicated as a product of fermentation [40]. 3-Hydroxypropanal (I-5), otherwise known as lactaldehyde, is naturally produced from glycerol and is predominant in all living organisms including humans and bacteria [41]. The molecule (I-5) links with lactic acid production during fermentation [42]. The emergence of lactic acids as a product of this fermentation reflects the type of fermentation that occurred. Lactic acid is associated with anaerobic fermentation [43].

An intermediate (I-3) in this study was one of the several derivatives of butanoic acid. The butanoic acid and the associated derivative were intermediates for fermentation [44,45,46]. This particular derivative (I-3) of butanoic acid was linked to fermentation for the first time in this study. The formation of reduced compounds such as lactic acid, ethanol, and butanol, among others, is evidence of the occurrence of metabolic pathways shifting from acidgenesis to solventogenesis [43]. One intermediate (I-1), a primary α-hydroxy ketone with butan-2-one substituted by a hydroxy group at positions 1 and 3 or a secondary α-hydroxy ketone, has a close structural analogue of 3,4-dihydroxybutan-2-one, whose phosphorylated form (l-3,4-dihydroxybutan-2-one 4-phosphate) is a precursor for the synthesis of riboflavin [47]. Formyl 7*E*-hexadecenoate, an effective nitrogen removal stimulant found in aquatic duckweed, was reported in lactic acid bacteria enhanced fermented milk [48,49]. Methyl petroselinate (C5) was also previously detected as fermented products of Korean red pepper and the extract of miswak plant, *Salvadora persica* L [50,51]. Although the study most probably provides for the first time the transformation of methyl petroselinate (C5) to formyl 7*E*-hexadecenoate (P2) during fermentation (Figure 2). This compound (P2), a component of the hexadecenoates derivatives identified in natural products, is a precursor for oleic acid biosynthesis [52,53].

This finding substantiates the report of Olawole, Okundigie, Rotimi, Okwumabua, and Afolabi [15], stating that fermented processed foods are chemically modified and nutritionally different from their raw sources. As stated earlier, fermentation is a metabolic reaction resulting from the activities of microorganisms that serve as agents for bio-transforming a particular compound to another. Fortunately, most of these biotransformations convert toxic compounds to safe and biologically active molecules [17,18,54]. This study gives credence to the potential of natural fermentation in improving the unique quality of agro-food crops, and the campaign targeting the detoxification of foods facilitated by probiotic lactic acid bacteria (LAB).

Notably, lactic acid was the primary compound discovered in the fermented extract. The production of lactic acid by lactic acid bacteria during fermentation has been attributed to be one of the reasons for the increased preservation of fermented foods. The lactic acid produced along with other organic acids during fermentation has been confirmed to inhibit the growth of food spoilage microorganisms [14,55]. This ability to hinder the growth of microorganisms is a reason why lactic acid bacteria are majorly used as probiotics. The lactic acid, thus detected in our fermented extract (Table 2), also suggests the presence of lactic acid bacteria during the fermentation, which could be isolated and further used as probiotics.

The presence of lactic acid bacteria in most fermented retailed foods was previously reported, making them excellent sources of live lactic acid bacteria including probiotics that provide health benefits to humans [18]. The compounds that were eliminated in our fermented extract are likely substrates for the growth of these probiotic organisms. This finding further substantiates the fermentation as a bioengineering tool to remove undesired compounds from harmful food crops as well as to modify their sensory properties such as taste, appearance, texture, and aroma that is derived from consuming the food product for safe consumption [13,15,17]. These changes in the physicochemical properties of such fermented foods could have been due to the possible growth of the three isolated LAB (*Lactococcus raffinolactis*, *Lactococcus lactis*, and *Pediococcus pentosaceus*).

The metabolized molecules that were components of the raw kernel (Table 1) of the *C. albidum* may constitute the essential nutrient molecules to support the growth of the probiotic organisms facilitating the fermentation. The seed of the *C. albidum* has carbohydrate (77.76%), moisture (7.5%), crude fiber (4.90%), proteins (4.22%), ash (3.01%), and fats (2.20%) as the primary nutrition components [56]. A study on the fatty acid profile of oil extracted from the seeds of *C. albidum* with gas chromatography-mass spectrometry (GC/MS) analysis presented n-hexadecanoic acid, 13-hexyloxacyctri-dec-10-en-2-one, oleic acids, octadecanoic acid, hexadecanoic acid, undecylenic acid, 9-octadecanal, and 9, 17-octadecadienal as components of the oil [57].

The pyrazine compounds are receiving increasing attention because they possess unique organoleptic properties. This increasing attention has led to the discovery of several pyrazine compounds in foods. Several others have also been extracted or artificially synthesized, and promoted as organoleptic and flavor enhancing agents [17,58]. The 2,5-dimethylpyrazine was detected in both the fermented and the unfermented seed. The presence of these pyrazine derivatives could be due to Maillard reactions in the seed [59,60]. Compounds in these groups (pyrazines and pyridine) are known to give that characteristic pleasant smell of peanut.

## 4. Materials and Methods

### 4.1. Plant Collection and Identification

The wholesome and fresh *C. albidum* fruits were procured from a local market commonly called “Oja Ota” in Ota, Ogun State, Nigeria (6.6841° N, 3.2153° E). The plant (Voucher number: FHI 112796) was identified and authenticated by the Forest Research Institute of Nigeria (FRIN), Ibadan, Nigeria. Additionally, an ethical permit (CHREC/49/2020) was obtained from the Covenant Health Research Institute, Canaanland, Ota, Nigeria, for the research to proceed.

### 4.2. Preparation of Flour from Kernel

The seeds were dehulled manually to remove the kernels of the fruits. The kernels were dried with the aid of an oven at 40 °C for about 24 h until it was crunchy and crispy enough to be ground into a fine powder. The dried kernels obtained were afterward ground into a fine powder with the aid of an electronic homogenizer. A powder extract was collected from the kernels of both the fermented and the raw-unfermented seeds. The extract from the raw-unfermented seeds was to serve as the control facilitating the identification of the uniquely metabolized bioactives that may be due to the fermentation process.

### 4.3. Production of Both the Unfermented and the Fermented Aqueous Extracts

The ground powder from the kernels was steeped in distilled water (1:3 *w*/*v*), stirred together, and allowed to stand for 24 h, under ambient temperature (26–33 °C). After 24 h, the solution was filtered with the use of filter paper (Whatman no. 1). The filtrate obtained was divided into two portions (A and B). The first portion (A) was subjected to fermentation before analysis; and the second portion (B) was directly used for further investigation of the raw-unfermented sample. The first portion (A) of the filtrate obtained was transferred into an airtight container and was left to naturally ferment anaerobically for 72 h under ambient temperature (26–33 °C). The second portion (B) of the filtrate obtained was immediately transferred into the freezer (−20 °C) to facilitate the freezing of the sample and to prevent it from undergoing fermentation. Both filtrates (A and B) were then removed from the freezer and transferred into a Telstar Cryodos-50 freeze dryer for the sublimation of the aqueous solvent until the dried extract remained. The dried extract from filtrates A and B were then labelled as the fermented aqueous extract and the unfermented aqueous extract, respectively.

### 4.4. Identification of the Isolated Organisms Facilitating the Fermentation

#### 4.4.1. Isolation of Lactic Acid Bacteria from the Fermented *C. albidum* Seeds

An aliquot of 20 µL of the fermented *C. albidum* solution was transferred with the aid of a sterile nichome inoculating loop into sterile de Man Rogosa Sharpe (MRS) agar (Oxoid medium) plates previously prepared according to the manufacturer’s instruction. This was thereafter streaked to observe the growth of distinct bacteria colonies. The plates were incubated microaerophilically using microaerophilic gas packs (Thermo scientific ltd., Basingstoke, UK) at 37 °C for 48 h. After incubation, single isolated colonies were picked with a sterile inoculating loop and cultured by repeated streaking on sterile MRS agar plates until pure cultures of the isolates were obtained.

#### 4.4.2. Gram Staining and Catalase Test

Isolates were sub-cultured on sterile MRS agar plates and incubated under microaerophilic conditions at 37 °C for 24 h. The young cultures were smeared on slides and Gram-stained to observe for shape and Gram-reaction. For the catalase test, the isolates were subjected to a reaction with hydrogen peroxide (H_2_O_2_); each isolate was picked and placed on a slide, and a drop of hydrogen peroxide (H_2_O_2_) solution (20 µL) was placed on each of them to observe for the formation of bubbles, where the bubble formation indicated a positive catalase reaction, while no bubble formation indicated a negative reaction.

#### 4.4.3. Biochemical Characterization of the Isolated Organisms

To identify the organisms, an Analytical Profile Index (API) 50 CHL Kit (Biomereux Ltd., Marcy-l’Etoile, France) for the identification of lactic acid bacteria was used to obtain the biochemical profile of the isolates; this was done according to the manufacturer’s instructions. The freshly sub-cultured 24 h LAB isolates were added into the API 50CHL basal medium and filled into wells on the API 50 CHL test strips. The strips were afterward incubated at 37 °C for 48 h and examined visually after 24 and 48 h of incubation. The indication of a positive or negative result was examined from the color change from a scale of 1 to 5, that is, from purple (1) to green (3) to yellow (5) at the 48 h-mark, and a value of >3 indicated a positive result. The results were afterward cross-referenced with the API^®^ databases with the use of the APIweb™ and based on the carbohydrate metabolism patterns obtained from the experiment, the names of the isolates were identified from the database.

### 4.5. Sample Extract Concentration for Gas Chromatography-Mass Spectrometry (GC-MS) Analysis

The remaining dried extracts (5 g) were re-dissolved in 25 mL of ethanol and was after that used for the GC-MS analysis.

#### Procedure for GC-MS Analysis

Shimadzu GC-MS (Model-QP2010SE, Osaka, Japan) equipped with a mass spectrometry detector was used for this experiment. The detector was set at an equilibrium time of 1.0 min, with an ion source and interface temperatures of 230 °C and 250 °C, respectively. The detector gain of 1.28 kV + 0.00 kV and a threshold of 2200 were attained during the experiment.

The sample (1.0 µL) was injected at 250 °C using a splitless mode into the GC-MS equipped with a pre-conditioned Optima 5 MS capillary column (30 × 0.25 mm) with 0.25 µL film thickness at 60 °C, and the sampling time was also set for 2.00 min. The flow control mode of the sample was set at linear velocity, and a pressure of 144.4 kPa was used to derive a total flow of 22.6 mL/min. The flow within the column used was 3.22 mL/min at a linear velocity of 46.3 cm/s. The purge flow of 3.0 mL/min and a split ratio of 5.1 was used. The oven temperature was programmed for 60, 120, and 290 °C at a holding time of 2, 2, and 3 min, respectively, while the flow rate of 15 was attained during the last two ovens’ programmed conditions. The NIST Mass Spectral Search Program for the NIST/EPA/NIH Mass Spectral Library (version 2.0 g, Gaithersburg, MD, USA) was used to perform the tentative identification. The common or scientific name of the identified compounds were obtained from the PubChem, Human metabolome, or yeast metabolome databases. The compounds detected by the tentative identification, with the percentage of coincidence (≥85%) were considered for the purpose of this study. Hence, all the compounds with either % area that was <1%, or with similarity index (≤85%) were manually excluded from the chromatographs (Table 2 and Table 3) since the GC-MS was in scan mode during the analysis. The proposed chemical reaction mechanisms were drawn with ChemAxon MarvinSketch software (15.9.14, Budapest, Hungary).

## 5. Conclusions

This study created two new bioactive compounds with immense therapeutic uses, the potential to enhance aroma, and the sweet taste of foods with the aid of natural fermentation of the kernel from the seeds of the *C. albidum*. Three possible phytochemicals with possible potential to support the growth of probiotics were identified. *Lactococcus raffinolactis*, *Lactococcus lactis*, and *Pediococcus pentosaceus* are the isolated health beneficial lactic acid bacteria promoting the fermentation of the seeds of the *C. albidum* for their probiotic characteristics. These strains of probiotics identified during this fermentation could, in the future, serve as a starter culture for the growth of lactic acid bacteria in the food industry [19]. The need to better identify the specific role of the individual identified phytochemicals serving as a possible starter culture for the growth of lactic acid bacteria should be critically studied. It is also suggested to identify the specific fermentation organism that metabolized each of the consumed phytochemicals and to link such metabolism to a specific phytochemical product(s) identified in this study. This understanding should help to biosynthesize these compounds in the future. The application of LAB isolates is recommended for incorporation into functional food products.

## Figures and Tables

**Figure 1 molecules-25-06021-f001:**
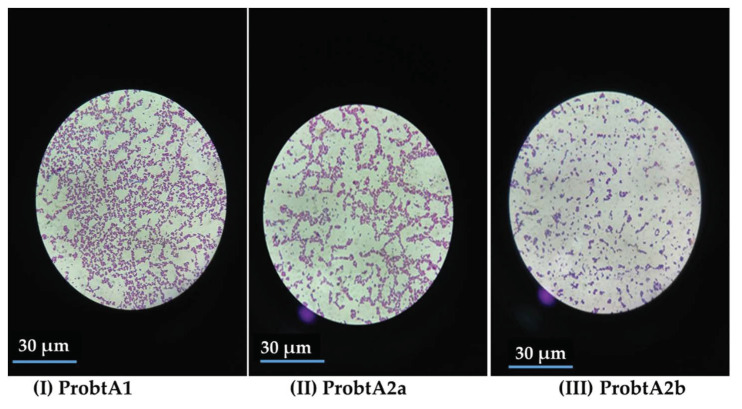
The microscopic view of the isolated lactic acid bacteria (LAB).

**Figure 2 molecules-25-06021-f002:**
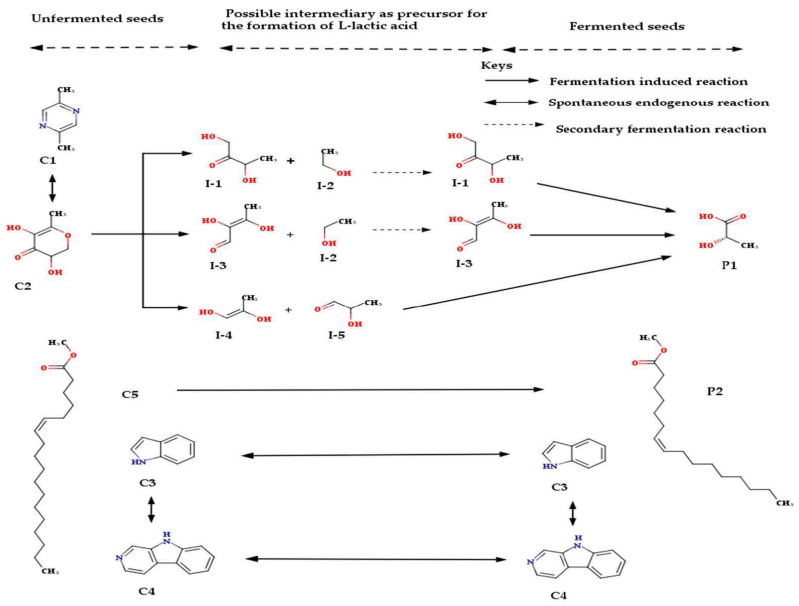
The proposed transformation mechanisms for the major molecules identified in the unfermented kernels of *C. albidum* during fermentation. Keys: C1: 2,5-dimethylpyrazine; C2: 3,5-dihydroxy-6-methyl-2,3-dihydro-4*H*-pyran-4-one; C3: Indole; C4: Norharman; C5: Methyl petroselinate; P1: l-Lactic acid; P2: formyl 7*E*-hexadecenoate; I-1: 1,3-dihydroxybutan-2-one; I-2: Ethanol; I-3: (2*E*)-2,3-dihydroxybut-2-enal; I-4: (1*E*)-prop-1-ene-1,2-diol; I-5: 2-hydroxypropanal.

**Table 1 molecules-25-06021-t001:** Compounds present in the gas chromatography-mass spectrometry (GC-MS) analysis of the unfermented aqueous extract of *C. albidum*.

S/N	Peaks	tR	Area (%)	Similarity Index (%)	Class of Compound	IUPAC Name	Common Name
1	4	5.74	1.27	89	Pyrazine	2,5-dimethylpyrazine	2,5-dimethylpyrazine
2	15	9.13	2.90	86	Ketone	3,5-Dihydroxy-6-methyl-2,3-dihydro-4*H*-pyran-4-one	2,3-Dihydro-3,5-dihydroxy-6-methyl-4*H*-pyran-4-one
3	19	11.25	1.31	91	Indole	1*H*-indole	Indole
4	41	17.69	3.01	87	Beta-carboline	9*H*-Pyrido[3,4-b]indole	Norharman
5	43	18.04	4.33	85	Fatty acid ester	Methyl cis-6-octadecenoate	Methyl Petroselinate

**Table 2 molecules-25-06021-t002:** Compounds present in the GC-MS analysis of the aqueous extract of fermented *C. albidum*.

S/N	Peaks	Tr	Area (%)	Similarity Index (%)	Class of Compound	IUPAC Name	Common Name
1	3	6.85	28.57	94	Carboxylic acid	(2*S*)-2-hydroxypropanoic acid	l-Lactic acid
2	6	11.21	1.51	92	Indole	1*H*-indole	Indole
3	20	17.71	5.07	95	Beta-carboline	9*H*-Pyrido[3,4-b]indole	Norharman
4	22	18.04	1.73	85	Fatty acid esters	7-Hexadecenoic acid, methyl ester, (*Z*)-	Formyl 7*E*-hexadecenoate

**Table 3 molecules-25-06021-t003:** The sugar metabolism pattern of the isolated lactic acid bacteria (LAB).

S/N	Substrates	ProbtA1	ProbtA2a	ProbtA2b
0	Control	−	−	−
1	Glycerol	−	−	−
2	Erythritol	−	−	−
3	d-Arabinose	−	−	−
4	l-Arabinose	+	+	+
5	d-Ribose	+	+	+
6	d-Xylose	−	−	−
7	l-Xylose	−	−	−
8	d-Adonitol	−	−	−
9	Methyl-βd-xylopyranoside	−	−	−
11	d-Galactose	+	+	+
12	d-Glucose	+	+	+
13	d-Fructose	+	+	+
14	d-Mannose	+	+	+
15	l-Sorbose	−	−	−
16	l-Rhamnose	−	−	−
17	Dulcitol	−	−	−
18	Inositol	−	−	−
19	d-Mannitol	−	+	+
20	d-Sorbitol	−	−	−
21	Methyl-αd-Mannopyranoside	−	−	−
22	Methyl-αd-Glucopyranoside	−	−	−
23	*N*-Acetylglucosamine	+	+	+
24	Amygdalin	−	−	−
25	Arbutin	+	+	+
28	Esculin ferric citrate	+	+	+
29	Salicin	+	+	+
30	d-Cellobiose	+	+	+
31	d-Maltose	+	+	+
32	d-Lactose	−	−	−
33	d-Mellibiose	+	+	+
34	d-Saccharose	+	+	+
35	d-Trehalose	+	+	+
38	Inulin	−	−	−
39	d-Melezitose	−	-	−
40	d-Raffinose	+	+	+
41	Amidon	+	−	−
43	Glycogen	−	−	−
44	Xylitol	−	−	−
45	Gentiobiose	+	−	+
46	d-Turanose	−	−	−
47	d-Lyxose	−	−	−
48	d-Tagatose	−	−	+
49	d-Fucose	−	−	−

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
