# Peer review of "Fermentation Enhanced Biotransformation of Compounds in the Kernel of Chrysophyllum albidum"

_molecules, 2020, doi:10.3390/molecules25246021_

Round 1

Reviewer 1 Report

The manuscript reports a study on the characterization of fermented and unfermented

 kernels of Chrysophyllum albidum. It could be interesting but some discussions should be implemented.

Introduction

The introduction part should be better focus on the aim of the research described into manuscript. For example, the general dissertation on bioactive compounds in foods (lines 50-59) should be omitted and substituted by a more specific presentation on the manuscript’s topic.

At lines 71-75, some sentences are reported about lactic acid bacteria. These observations are well known and with a wide field of applications. I suggest authors relate with the fermentation applied in the described or similar studies.

Furthermore, from the introduction it was not directly reported that there were not previous researches about fermentation of Chrysophyllum albidum (fruit or kernel). In other manuscript parts, authors reported that their study was the first about this topic, but they did not underline it into the introduction. Neither studies about the product characterization were reported, also for this observation authors have to consider my previous comment. Finally, authors should underline the goal of the study. Is it a simple characterization or it would improve knowledge about nutritional value of the product obtained? I think the last is the real goal of authors.

I suggest rewriting and implementing introduction with a better discussion about the goal (s) of the study.

Results

Authors should check the IUPAC names, common names and keys of the molecules reported in figures. I didn’t find the molecule 1-II, 2,5-methano-2H-furo[3,2-b]pyran, hexahydro- (figure 1), in tables. Authors discussed on it at lines 136-140 as a significant molecule. Other molecule names should check also. I think there should be a correspondence between names in figures, text and tables.

Discussion

The discussion at lines 152-163 is partly confuse, some aspects were discussed without a linear relation. Authors reported the effect of acidic medium due to fermentation, presence of rizatriptan and a general discussion about fermentation effects (particularly at lines 161-163). I suggest organizing better this part focusing on the goal and subject matter of the presented study.

In the discussion part about lactic acid production and lactic acid bacteria (lines 165-177) authors repeated the benefits of lactic acid bacteria. They should organize better this part also.

At line 179, the phrase “to modify their aesthetic properties for safe consumption” is not clear. In particular, it is not explained what authors mean with “aesthetic properties for safe consumption”.

The phrase at lines 186-187 appears incomplete: in this….

At lines 191, authors should complete the phrase. I think they would mean the free fatty acids.

At lines 201-206, authors discussed about pyrazynes. Did they verify if these compounds were formed as Maillard derivatives during the drying process of kernels?

Conclusions.

This part should be improve. At the beginning of conclusions, authors stated that “the study created bioactive compounds”, actually the fermentation produces bioactive compounds.

Material and methods

Authors should better describe the production of both the unfermented and the fermented aqueous extracts. Particularly they should describe extraction and fermentation process. At least a total mesophilic count together with lactic acid bacteria enumeration could be useful to describe the process.

Author Response

We have attended to the comments raised by the reviewers. The comments were indeed beneficial and constructive. It has improved the quality of the manuscript, and we are very grateful for the contributions. Kindly find below our point by point response to each of the comments.

COMMENT: The manuscript reports a study on the characterization of fermented and unfermented kernels of Chrysophyllum albidum. It could be interesting but some discussions should be implemented.

RESPONSE: We are grateful for the positive comment. The suggestion had been effected, and we affirm that it has made the study interesting.

COMMENT: The introduction part should be better focus on the aim of the research described into manuscript. For example, the general dissertation on bioactive compounds in foods (lines 50-59) should be omitted and substituted by a more specific presentation on the manuscript’s topic.

RESPONSE: We have added the sentence ‘These efforts should further provide an understanding of the nature of biotransformation of such identified molecules during the fermentation, and the possible health benefits of the product of the fermentation.’ to the objective so to make the objective clearer and to relate well with the title (See line 102-104 in the revised manuscript). The term ‘bioactive’ in the previous manuscript was changed to phytochemical (See line 101). More statements was also included as line 74-94 in the revised manuscript to relate the article more to the aims of the manuscript.

COMMENT: At lines 71-75, some sentences are reported about lactic acid bacteria. These observations are well known and with a wide field of applications. I suggest authors relate with the fermentation applied in the described or similar studies.

RESPONSE: The intention of the reviewer is not clear to us. We had further related the activities of the organisms to the natural fermentation used for this experimentation. See line 74-94 of the revised manuscript.

COMMENT: Furthermore, from the introduction it was not directly reported that there were not previous researches about fermentation of Chrysophyllum albidum (fruit or kernel). In other manuscript parts, authors reported that their study was the first about this topic, but they did not underline it into the introduction. Neither studies about the product characterization were reported, also for this observation authors have to consider my previous comment. Finally, authors should underline the goal of the study. Is it a simple characterization or it would improve knowledge about nutritional value of the product obtained? I think the last is the real goal of authors.

I suggest rewriting and implementing introduction with a better discussion about the goal (s) of the study.

RESPONSE: We have added an additional sentence ‘These efforts should further provide a better understanding of the nature of biotransformation of such identified molecules during the fermentation, and the possible health benefits of the product of the fermentation.’ to the objective so to make the objective clearer, and to relate well with the title (See line 101-104 in the revised manuscript). The term ‘bioactive’ in the previously stated objective was also changed to phytochemical (See line 101).

Results

COMMENT: Authors should check the IUPAC names, common names and keys of the molecules reported in figures. I didn’t find the molecule 1-II, 2,5-methano-2H-furo[3,2-b]pyran, hexahydro- (figure 1), in tables.

RESPONSE: ‘2,7-dioxatricyclo[4.3.1.03,8]decane’, which is another form of the scientific name from PubChem, Human metabolome, or yeast metabolome databases, had been replaced with the ‘2,5-methano-2H-furo[3,2-b]pyran, hexahydro-’ that was sourced directly from the GC-MS database used. Also, the IUPAC name for each of the molecule identified with peak 5, 15, 18, 23, 27, 28, 32, 34, 43, 44, 46, 49-52, 54, and 59 in table 1 were replaced with a similar but more appropriate IUPAC name sourced directly from the GC-MS database used. Other molecules had been rechecked and confirmed, as suggested. The molecules identified with peak 5, 10, 18, 22, 25, 31, and 38 were each similarly replaced with another IUPAC name that is sourced directly from the GC-MS database used.

Discussion

COMMENT: The discussion at lines 152-163 is partly confuse, some aspects were discussed without a linear relation. Authors reported the effect of acidic medium due to fermentation, presence of rizatriptan and a general discussion about fermentation effects (particularly at lines 161-163). I suggest organizing better this part focusing on the goal and subject matter of the presented study.

RESPONSE: i.) Amine was excluded from, and the imidazole and the oxazole were added to the classes of compounds that reduced during the fermentation.

ii.) The amine and the tryptamine were added to the chemical groups that did not change during the fermentation.

iii.) the statement ‘Likely, the consumption of both the raw and the fermented kernels of C. albidum may relieve pain since they both contain rizatriptan that was earlier implicated in the treatment of migraine [19,20]’ was moved from line 161-162 of the submitted manuscript to line 244-246 of the revised manuscript, to enhance the flow of the paragraph.

COMMENT: In the discussion part about lactic acid production and lactic acid bacteria (lines 165-177) authors repeated the benefits of lactic acid bacteria. They should organize better this part also.

RESPONSE: The second statement, ‘The ability to produce organic acid that can hinder the growth of microorganisms, is a reason why lactic acid bacteria …….’ was to refer to the previous statement. It has been corrected as ‘This ability to hinder the growth of microorganisms is a reason why lactic acid bacteria ….. (Table 2)’ for better understanding (See line 197-199 in the revised manuscript).

COMMENT: At line 179, the phrase “to modify their aesthetic properties for safe consumption” is not clear. In particular, it is not explained what authors mean with “aesthetic properties for safe consumption”.

RESPONSE: aesthetic properties, we mean the sensory feelings, such as taste, appearance, texture, and aroma, that is derived from consuming the food product. The meaning of the statement ‘their aesthetic properties for safe consumption’ has been adjusted to ‘their sensory properties, such as taste, appearance, texture, and aroma, that is derived from consuming the food product for safe consumption’ for improved understanding (see line 210-211, in the revised manuscript).

COMMENT: The phrase at lines 186-187 appears incomplete: in this….

RESPONSE: The comment is not clear since the stated line 186-187 is pointing to Figure 2 in the submitted manuscript. We assume the reviewer refers to the statement ‘The process of fermentation used in this could have metabolized the DL-Tryptophan, which is an aromatic amino acid.’ in line 237 – 238. This statement had been corrected as ‘The process of fermentation used in this study could have metabolized the DL-Tryptophan, which is an aromatic amino acid.’ in line 212-214 in the revised manuscript.

COMMENT: At lines 191, authors should complete the phrase. I think they would mean the free fatty acids.

RESPONSE: It is not clear because the line 191 refer to figure 2 in the submitted manuscript. We, therefore, assume the reviewer referred to the statement ‘Fatty acids are usually not found in living organisms.’ in line 242 of the submitted manuscript. The fatty acids had been changed to the suggested ‘free fatty acids’ in line 223 of the revised manuscript for better understanding.

COMMENT: At lines 201-206, authors discussed about pyrazynes. Did they verify if these compounds were formed as Maillard derivatives during the drying process of kernels?

RESPONSE: This suggestion is out of the scope of the study since maillard reaction is not covered in this study. However, a new statement ‘The presence of these pyrazine derivatives could be due to maillard reactions in the seed [28,29].’ was incorporated line 239-240 of the revised manuscript to give account for the presence of pyrazine.

Conclusions.

COMMENT: This part should be improve. At the beginning of conclusions, authors stated that “the study created bioactive compounds”, actually the fermentation produces bioactive compounds.

RESPONSE: the phrase ‘Seven new’ was incorporated to the bioactive compounds that were stated at the beginning of the conclusion to be more precise and to differentiate the findings of this study from other reports on fermentation producing bioactive compounds. Also, a new statement, ‘Six possible phytochemicals with possible potential to support the growth of probiotics were identified.’ was added to the conclusion for more emphasis on the contribution of the study (See line 257-258).

Material and methods

COMMENT: Authors should better describe the production of both the unfermented and the fermented aqueous extracts. Particularly they should describe extraction and fermentation process. At least a total mesophilic count together with lactic acid bacteria enumeration could be useful to describe the process.

RESPONSE: (i.) we have improved the understanding of the production process by ensuring that there is consistency with the meaning of the first (A) and the second (B) portions (See line 290-293). Another statement, ‘The dried extract from filtrates A and B were then labeled as the fermented aqueous extract and the unfermented aqueous extract respectively’ was also included (see line 296-297) for more clarity.

(ii). Total mesophilic count: No action taken since this suggested experimentation is outside the scope of this study. It was also stated in the conclusion that such work, like characterisation and sequencing, the gene of the identified organisms is on-going.

Reviewer 2 Report

Dear authors

You must improve the quality of your manuscript, take into account the observations and make changes:

Title

Line 2: It says “fermentation enhanced biotransformation…”, but in the document it is not evidente, the improved process is not found. What was the improved process?

Abstract

Line 18: Indicates microorganisms associated with fermentation but its not evident in the document. What are the microorganisms associated with the fermentation process?

Introduction

Line 41: You are using the term “food security”, it lacks context and the sentence loses its meaning. Lines 76 a 78: There is not correlation between the title and the objective of the work.

Results

You must using the notation given for chromatography: for retention time it is "tR". The retention times are writting to 2 decimal places. Its make the changes to the document (text, tables, etc). Today, those of us who work in chromatography use the term "tentative identification" by GC / MS.

In the tables you must include the % match with the database. Review the IUPAC and common names of compounds. It should include the chromatograms and improve the resolution of the images.

Line 124: Because it says that the molecules were catabolized by probiotics: Did it prove it? Was it carried out in the study to prove it?

Discussion

You must include a paragraph with the strengths and weaknesses of your work.

Line 185: Possible growth of Lactobacillus plantarum: Did you isolate, identify and characterize the microorganism? and other species?

Line 199: What are the components of the kernel of C. albidum?

Conclusion

You must rewrite again. The conclusions correspond to the work done, they are not recommendations or work activities to be done in the future. Methods What was the percentage of agreement of the mass spectra obtained with the mass spectra of the database. What was the MS database that you used to perform the tentative identification of the compounds?

Lines 248 to 249: How can you showed that it corresponds to a fermentation-type process by microorganisms?

Line 256: Did you worked with a mass detector?, the acronym indicates another type of detector. Which was or were the detectors used?

Line 260: Rewrite the magnitude of the temperature in whole numbers.

Author Response

We have attended to the comments raised by the reviewers. The comments were indeed beneficial and constructive. It has improved the quality of the manuscript, and we are very grateful for the contributions. Kindly find below our point by point response to each of the comments.

REVIEWER- 2

Title

COMMENT: Line 2: It says “fermentation enhanced biotransformation…”, but in the document it is not evidente, the improved process is not found. What was the improved process?

RESPONSE: The title of the manuscript was not changed since the title clearly depicts the content of the document. The phrase, ‘fermentation enhanced biotransformation’ in the title suggests that the fermentation process described in line 284-297 in the revised manuscript was used to facilitate the biotransformation of the phytochemicals in the unfermented extracts (Table 1) to different types of phytochemicals in the fermented extracts (Table 2). The notable phytochemicals that were biotransformed were also depicted in Figure 2 and Figure 3.

Abstract

COMMENT: Line 18: Indicates microorganisms associated with fermentation but its not evident in the document. What are the microorganisms associated with the fermentation process?

RESPONSE: The stated line 18 was not part of the results of the abstract, and the statement is satisfactory since the role of microorganisms (Lactic acid bacteria) had already been established to be the cause of fermentation (Already stated in line 49-52, and line 63-69 of the submitted manuscript as well as the line 50-54, and line 63-98 of the revised manuscript). In the presence of appropriate substrate and conditions, the microorganisms are the living organisms facilitating the biochemical processes inducing the fermentation in food (See line 64-70 of the submitted manuscript). Also, we have identified the microorganisms involved, and the extensive characterisation for a possible novel lactic acid bacteria is in progress (Unpublished report). This statement was well stated in our conclusion (see line 272-275 of the submitted manuscript and line 258-260 of the revised manuscript).

Introduction

COMMENT: Line 41: You are using the term “food security”, it lacks context and the sentence loses its meaning. Lines 76 a 78:

RESPONSE: We believe the issue raised has to do with the flow of the statement since the sentence clearly communicates our intention. Therefore, the statement, ‘These non-edible seeds may have the potentials to be transformed into useful purpose(s). Efforts are on the increase to reveal the potentials and benefits of some of these non-edible seeds since they may be transformed into useful purpose(s).’ was added to properly connect with the sentence preceding it and for better understanding (See line 40-42 of the revised manuscript).

COMMENT: There is not correlation between the title and the objective of the work.

RESPONSE: We have added an additional sentence ‘These efforts should further provide an understanding of the nature of biotransformation of such identified molecules during the fermentation, and the possible health benefits of the product of the fermentation.’ to the objective so to make the objective clearer, and to relate well with the title (See line 101-104 in the revised manuscript). The term ‘bioactive’ in the previously stated objective was also changed to phytochemical (See line 101 of the revised manuscript).

Results

COMMENT: You must using the notation given for chromatography: for retention time it is "tR". The retention times are writting to 2 decimal places. Its make the changes to the document (text, tables, etc). Today, those of us who work in chromatography use the term "tentative identification" by GC / MS.

RESPONSE: ‘RT’ has been changed to ‘tR’, and all the figures for the retention time were changed to 2 decimal places as informed.

COMMENT: In the tables you must include the % match with the database. Review the IUPAC and common names of compounds.

RESPONSE: A column for the percentage similarity index generated during the experimentation had been created and incorporated into Table 1 and Table 2.

COMMENT: It should include the chromatograms and improve the resolution of the images.

RESPONSE: The resolution of the images was as specified by the Journal. The submission system screened it before the figures were accepted.

COMMENT: Line 124: Because it says that the molecules were catabolized by probiotics: Did it prove it? Was it carried out in the study to prove it?

RESPONSE: Yes, we did not experiment the stated catabolism, which may also be an anabolic reaction. Therefore, the term ‘catabolised’ had been changed to ‘metabolised’ since it has not been proven that the stated phytochemicals were break-down into simpler molecules (See line 30 and line 265 of the revised manuscript). The same word was further changed to ‘consume’ in line 139 of the revised manuscript.

Discussion

COMMENT: You must include a paragraph with the strengths and weaknesses of your work.

RESPONSE: The strength of this work is the significant findings earlier stated in the manuscript. Also, the deficiency of this work was also suggested as areas for further studies in line 262-267 of the revised manuscript. Additionally, natural fermentation provides an opportunity for a better overview of the biochemical and microbial activities during the domestic preparation of fermented foods. On the other hand, the limitation of this natural or spontaneous type of fermentation is the high risks of spoiling microbial communities of foods and the domination of microbial strains that could be dangerous for human health (Capozzi et al., 2017). The latter statement was added to the revised manuscript (line 92-95).

COMMENT: Line 185: Possible growth of Lactobacillus plantarum: Did you isolate, identify and characterize the microorganism? and other species?

RESPONSE: The stated growth of Lactobacillus plantarum in line 236 of the submitted manuscript was as earlier reported in the reference provided (Reference 15, Goswami et al., 2018).  Also, it was mentioned in the conclusion that the identification is on-going (see line 272-275 of the submitted manuscript and line 258-260 of the revised manuscript).

COMMENT: Line 199: What are the components of the kernel of C. albidum?

RESPONSE: The statement refers to the components in this study. The appropriate table 1 had been added to the statement for more clarity (See line 231 of the revised manuscript).

Conclusion

COMMENT: You must rewrite again. The conclusions correspond to the work done, there are no recommendations or work activities to be done in the future.

RESPONSE: The statement ‘The need to identify the specific role of the individual identified phytochemicals serving as a possible starter culture for the growth of lactic acid bacteria should be critically studied. It is also recommended to identify the specific fermentation organism that metabolised each of the consumed phytochemicals and to link such metabolism to a specific phytochemical product(s) identified in this study. These understanding should help to biosynthesise those compounds in the future.’ had been included in the conclusion as a recommendation for further studies as suggested (See line 262-267 of the revised manuscript).

COMMENT: Methods What was the percentage of agreement of the mass spectra obtained with the mass spectra of the database. What was the MS database that you used to perform the tentative identification of the compounds?

RESPONSE: The NIST Mass Spectral Search Program for the NIST/EPA/NIH Mass Spectral Library (version 2.0 g, USA) was used to perform the tentative identification. This statement has been added to line 313-316 of the revised manuscript for better understanding.

COMMENT: Lines 248 to 249: How can you showed that it corresponds to a fermentation-type process by microorganisms?

RESPONSE: The intention of the reviewer is not clear since the stated lines is not related to microorganisms or fermentation-type process. We suspect the issue must be on the stated growth of Lactobacillus plantarum in line 236 of the submitted manuscript, which was from a report in the reference provided (Reference 15, Goswami et al., 2018).  It was also mentioned in the conclusion that the identification is on-going (see line 272-275 of the submitted manuscript and line 258-260 of the revised manuscript).

COMMENT: Line 256: Did you worked with a mass detector?, the acronym indicates another type of detector. Which was or were the detectors used?

RESPONSE: Yes, a mass detector was used for the experimentation. The FTD has been changed to ‘mass spectrometry detector’ in line 302 of the revised manuscript.

COMMENT: Line 260: Rewrite the magnitude of the temperature in whole numbers.

RESPONSE: Action was taken. All the temperature had been changed to whole numbers in the line 306-307, and line 311 of the revised manuscript.

Round 2

Reviewer 1 Report

The changes applied and the answers to my observations improve the manuscript and its comprehension.

I am sorry I have not clearly put my observation about LAB discussed in the introduction part. I have suggested to authors to simplify and report only applications on similar research (if there are). The introduction discussed about LAB in a general way. This is one of the less important observation.

My observation at phrase of lines 186-187 (1st manuscript) referred to a lexical difficulty. Authors have specified the term study in the phrase and now it is clear; this could improve the legibility of this part. Probably could be differences on the lines numeration.

Author Response

Dear Reviewer,

We have attended to the comments raised by the reviewers. The comments were indeed beneficial and constructive. It has improved the quality of the manuscript. Authors appreciate the reviewer process, which drove us to the best of our potentials. Kindly find below our point by point response to each of the comments.

Reviewer 1

Comments and Suggestions for Authors

The changes applied and the answers to my observations improve the manuscript and its comprehension.

COMMENT: I am sorry I have not clearly put my observation about LAB discussed in the introduction part. I have suggested to authors to simplify and report only applications on similar research (if there are). The introduction discussed about LAB in a general way. This is one of the less important observation.

Response: We have included the requested detailed on the isolated LAB (See line 14-15, 20-26, 30-31 (abstract), 109-117 (results), 141-149 (discussion), and 286-315 (methodology) in the revised manuscript.

COMMENT: My observation at phrase of lines 186-187 (1st manuscript) referred to a lexical difficulty. Authors have specified the term study in the phrase and now it is clear; this could improve the legibility of this part. Probably could be differences on the lines numeration.

Response: We (authors) expressed our appreciation to the suggestion that led to this stated clarity and improved legibility.

Reviewer 2 Report

Dear Authors: Below you will find comments and observations made on your manuscript

TITLE
The arguments presented are not strong to validate the title.

SUMMARY
The arguments presented are not strong. What microorganisms? Scientific names, at least gender.

INTRODUCTION
Lines 40 to 42: The seeds are not transformed, they are the compounds present in the seed that are transformed by the fermentation process. The precursor must be known and the transformation route described or proposed until reaching the product (biotransformed compound).
Lines 75 to 95: Ok. It is necessary to include names of microorganisms. Line 101: Ok. Line 101 to 104: Ok.

RESULTS
The tentative identification of the compounds is carried out from the coincidence percentages (GC / MS), the value of 85% is taken as a selection and identification criterion. You must make the changes. The chromatographic results are taken from criteria such as:% area (> 1% for major compounds),% coincidence (≥ 85%). It must include chromatograms, if it does not, it is suggested to reject the manuscript. Low resolution figures or images do not motivate reading and analyzing the manuscript.

DISCUSSION
Strengths and weaknesses are included in the discussion, in the form of paragraph (s). Line 258 to 260: Lactobacillum plantarum ?, not found. Line 231: What are the kernel components of C. albidum ?, not found.

CONCLUSION
This section concludes on what has been done at work (according to the methodology). They are not recommendations or suggestions.

METHODS
Okay. Include the exclusion criterion of the compounds by tentative identification, which is the percentage of coincidence ≥85%

Regards   Reviewer

Author Response

Dear Reviewer,

We have attended to the comments raised by the reviewers. The comments were indeed beneficial and constructive. It has improved the quality of the manuscript. Authors appreciate the reviewer process, which drove us to the best of our potentials. Kindly find below our point by point response to each of the comments.

Reviewer 2

Comments and Suggestions for Authors

TITLE
COMMENT: The arguments presented are not strong to validate the title.

Response: We have included the requested detailed on the isolated LAB (See line 14-15, 20-26, 30-31 (abstract), 109-117 (results), 141-149 (discussion), and 286-315 (methodology) in the revised manuscript to justify the title.

SUMMARY
COMMENT: The arguments presented are not strong. What microorganisms? Scientific names, at least gender.

Response: We have included the requested detailed on the isolated LAB (See line 14-15, 20-26, 30-31 (abstract), 109-117 (results), 141-149 (discussion), and 286-315 (methodology) in the revised manuscript.

INTRODUCTION
COMMENT: Lines 40 to 42: (i.) The seeds are not transformed, they are the compounds present in the seed that are transformed by the fermentation process. (ii.) The precursor must be known and the transformation route described or proposed until reaching the product (biotransformed compound).

Response: (i.) The fermentation also transform the physical properties of the seeds leading to other products from the process, while at the same time transforming the natural phytochemicals. This is evidence by the change in colour, texture and some other physical properties products (liquid pap from cereals like sorghum, nutritious milk from the seeds of soybeans or Adenanthera pavonina, flakes and tapioca from cassava tubers, food condiments like iru or maggi from African locust beans) derived from such fermentation process [1-7]. This statement had been included into line 40-46 in the revised manuscript to cover the gap in the literature provided.

(ii.) The proposed mechanisms indicating the precursor molecules had been provided as Figure 1.

COMMENT: Lines 75 to 95: Ok. It is necessary to include names of microorganisms. Line 101: Ok. Line 101 to 104: Ok.

Response: We have included the requested detailed on the isolated LAB (See line 14-15, 20-26, 30-31 (abstract), 109-117 (results), 141-149 (discussion), and 286-315 (methodology) in the revised manuscript.

RESULTS
COMMENT: The tentative identification of the compounds is carried out from the coincidence percentages (GC / MS), the value of 85% is taken as a selection and identification criterion. You must make the changes. The chromatographic results are taken from criteria such as:% area (> 1% for major compounds),% coincidence (≥ 85%). It must include chromatograms, if it does not, it is suggested to reject the manuscript.

Response: All the compounds with either % area that is <1%, or with similarity index (≤ 85%) manually excluded from the chromatographs (Table 1 and Table 2) since the GC-MS was in scan mode during the analysis. This statement had been included into the methods for the GC/MS analysis for clarity (See line 334-339). We did not implement the same into the chromatograms since the ‘scan mode’ was used for the GC-MS operation. This have been clearly indicated in the methodology (See line 334-339) as earlier indicated. The full tables are submitted as supplementary document for further consultation to the readers.

COMMENT: Low resolution figures or images do not motivate reading and analyzing the manuscript.

Response: The resolution of all the figures/images had been improved to 650 dpi.

DISCUSSION
COMMENT: Strengths and weaknesses are included in the discussion, in the form of paragraph (s). Line 258 to 260: Lactobacillum plantarum ?, not found. Line 231: What are the kernel components of C. albidum?, not found.

Response: The state organism ‘Lactobacillum plantarum’ from literature has been changed to ‘the three isolated LAB (Lactococcus raffinolactis, Lactococcus lactis, and Pediococcus pentosaceus)’ that was isolated in our study (See line 225 of the revised manuscript). Also, the essential molecules suggested to serve as nutrient to support the growth of the probiotic organisms facilitating the fermentation were as stated in Table 2. The other nutritional components of the C. albidum was included into line 228-233 of the revised manuscript.

CONCLUSION
COMMENT: This section concludes on what has been done at work (according to the methodology). They are not recommendations or suggestions.

Response: We further included recommendation from the study into line 255-256 of the revised manuscript. The recommendations on aspect that will cover the parts yet to be performed were stated in the submitted manuscript, and is presented in line 250-255 of the revised manuscript.

METHODS
COMMENT: Okay. Include the exclusion criterion of the compounds by tentative identification, which is the percentage of coincidence ≥85%.

Response: The statement ‘The compounds detected by the tentative identification, with the percentage of coincidence (≥85%) were considered for the purpose of this study.’ had been included into line 234-236 of the revised manuscript.